# XOL-1 regulates developmental timing by modulating the H3K9 landscape in *C. elegans* early embryos

**Eshna Jash**[iD]**, Anati Alyaa Azhar**[iD]**, Hector Mendoza**[iD]**, Zoey M. Tan**[iD]**, Halle Nicole Escher, Dalia S. Kaufman**[iD]**, Györgyi Csankovszki**[iD]*

Department of Molecular, Cellular, and Developmental Biology, University of Michigan, Ann Arbor, Michigan, United States of America

* gyorgyi@umich.edu

**Data Availability Statement:** RNA-seq datasets produced in this study are available at the NCBI GEO database under GSE262626. Scripts used to

## Abstract

Sex determination in the nematode *C. elegans* is controlled by the master regulator XOL-1 during embryogenesis. Expression of *xol-1* is dependent on the ratio of X chromosomes and autosomes, which differs between XX hermaphrodites and XO males. In males, *xol-1* is highly expressed and in hermaphrodites, *xol-1* is expressed at very low levels. XOL-1 activity is known to be critical for the proper development of *C. elegans* males, but its low expression was considered to be of minimal importance in the development of hermaphrodite embryos. Our study reveals that XOL-1 plays an important role as a regulator of developmental timing during hermaphrodite embryogenesis. Using a combination of imaging and bioinformatics techniques, we found that hermaphrodite embryos have an accelerated rate of cell division, as well as a more developmentally advanced transcriptional program when *xol-1* is lost. Further analyses reveal that XOL-1 is responsible for regulating the timing of initiation of dosage compensation on the X chromosomes, and the appropriate expression of sex-biased transcriptional programs in hermaphrodites. We found that *xol-1* mutant embryos overexpress the H3K9 methyltransferase MET-2 and have an altered H3K9me landscape. Some of these effects of the loss of *xol-1* gene were reversed by the loss of *met-2*. These findings demonstrate that XOL-1 plays an important role as a developmental regulator in embryos of both sexes, and that MET-2 acts as a downstream effector of XOL-1 activity in hermaphrodites.

## Author summary

Various organisms have differing ways of determining, at a molecular level, what the sex of a developing embryo is supposed to be. The two sexes in the nematode *C. elegans*, hermaphrodite and male, have different numbers of X chromosomes. Hermaphrodites have two X chromosomes and males only have one. This mismatch raises an additional problem as hermaphrodites will have twice the amount of genes expressed from the X compared to males. This is solved by a process called dosage compensation, which equalizes gene expression between the sexes. A molecular sensor called XOL-1 detects the number

process and analyze the data can be found at https://github.com/eshnaj/jash_xol1_dev_time_paper.

**Funding:** This work was supported by the National Science Foundation, grant number MCB1923206 (GC). The Youngman Fellowship at the University of Michigan provided partial support to EJ. Some strains were provided by the CGC, which is funded by NIH Office of Research Infrastructure Programs (P40 OD010440). The funders had no role in the study design, data collection and analysis, decision to publish, or preparation of the manuscript.

**Competing interests:** The authors have declared that no competing interests exist.

of X chromosomes in an embryo and kick-starts the process of proper sexual development and/or dosage compensation. XOL-1 was known to be very important for activating male development but was believed to not have any role in the development of the hermaphrodite sex. We show that XOL-1 has some crucial roles in hermaphrodite embryos in controlling the rate of development of the embryo, and regulating the timing of dosage compensation. We also show that MET-2, a protein that deposits repressive methyl marks on DNA-bound histone proteins, is involved in this process.

## Introduction

Embryogenesis is the sequential process of the development of an embryo from fertilization to birth or hatching. Several molecular mechanisms during early embryogenesis have major impacts on the formation and development of the embryo. Two of these are the processes of sex determination and, in some organisms, dosage compensation. Activation of the appropriate sex determination pathways in particular is a critical step in the developmental pathways that regulate sexual reproduction. Within the context of multicellular organisms that have heterogametic sexes, sex determination is an important step that is established very early during embryogenesis. In the case of the nematode *C. elegans*, activation of the appropriate sex determinations pathway in the two sexes is based on the number of X chromosomes in the developing embryo [1,2]. Hermaphrodites have two X chromosomes, whereas males only have a single X chromosome. Activation of the appropriate sex determination pathways in both hermaphrodites and males are important for the development of sexually dimorphic anatomy and sexually specialized cells. In *C. elegans*, hermaphrodites and males differ extensively in their appearance and behavior, and ~30–40% of somatic cells in adults are known to be sexually specialized [3]. Some of these sexually specialized somatic cells, such as sex-specific neuronal cells, arise during early embryogenesis [4].

In addition, hermaphrodite embryos also need to activate dosage compensation [5]. Dosage compensation is the process through which organisms with chromosome-based methods of sex determination equalize gene expression from sex chromosomes between the two sexes. Hermaphrodites have two X chromosomes and the males have only one. This poses a challenge since the X chromosomes contain many genes that are not involved specifically in sex determination. Organisms across the animal kingdom have significant divergence in their molecular mechanisms for achieving dosage compensation. In mammals, there is near complete inactivation of one of the two X chromosomes of the female to match the transcriptional output from the single X chromosome in males [6]. Other organisms such as *Drosophila* and birds have their own unique mechanisms to establish dosage compensation which involve hyperactivation of one of the sex chromosomes, though in the case of birds the hyperactivation is incomplete [7,8]. In the case of *C. elegans*, the transcriptional output of both the hermaphrodite X chromosomes is repressed by half to match that of the single male X chromosome through the activity of the dosage compensation complex (DCC) [5].

XOL-1 is the master switch that regulates both sex determination and dosage compensation in the *C. elegans* [9,10]. XOL-1 acts as a sensor that can be toggled on or off depending on the ratio of signal elements from the X chromosome and the autosomes [9]. These elements on the X chromosome such as *sex-1*, *sex-2*, *ceh-39* and *fox-1* are known as X signal elements (XSEs) and those on the autosomes such as *sea-1* and *sea-2* are known as autosome signal elements (ASEs) [1,11–16]. The ratio of these elements differs between the two *C. elegans* sexes. Hermaphrodites have two X chromosomes resulting in an X: A ratio of 1, whereas the males only

have one X chromosome resulting in an X:A ratio of 0.5. XSEs and ASEs compete to repress and activate expression of the *xol-1* gene in early embryos, respectively [15]. Due to the lower X:A ratio in males, transcription of XOL-1 is activated. XOL-1 then represses SDC-1, SDC-2 and SDC-3, which leads to the activation of the male sex determination pathway and the repression of hermaphrodite development and dosage compensation [1,9,10]. In hermaphrodites, due to the higher X:A ratio, the reverse occurs where XOL-1 is repressed. Hermaphrodite development and dosage compensation pathways are then activated, and male development is shut down [9].

While sex determination and dosage compensation are related pathways and are regulated by a common pathway during in the early stages of embryogenesis in *C. elegans*, the two pathways eventually diverge and can be regulated independently [10,17]. The SDC proteins influence both sex determination and dosage compensation, but the two pathways are independently regulated by the targets of the SDC proteins. Modulation of the SDC proteins can affect sex determination through their transcriptional regulation of the *her-1* gene [17]. In hermaphrodites, the SDC proteins repress *her-1*, a gene important for male development. *her-1* activates the male sex determination pathway, and represses the hermaphrodite sex determination pathway [18,19]. In addition to the repression of *her-1* in hermaphrodites, the SDC proteins also turn on dosage compensation by activating the DCC [17]. The DCC binds to recruitment sites on the X chromosome and spreads along the length of the chromosome to repress transcription [20–23]. The DCC and the additional ancillary pathways it recruits use multiple mechanisms to establish and maintain dosage compensation on the hermaphrodite X chromosomes throughout somatic tissues in larvae and adults [24–28]. These include X-specific deposition of H4K20me1 repressive mark by DPY-21 [27–30], tethering of H3K9me2/me3 regions to the nuclear lamina [25], contributions by the nuclear RNAi machinery [24], and the formation of TADs on the X chromosomes [26,31].

The initiation of dosage compensation in developing hermaphrodite embryos is linked to a loss in developmental plasticity and the onset of cellular differentiation programs [32,33]. Several post-translational histone modifications are known to contribute to both the processes of dosage compensation and the loss of developmental plasticity. One of these are the H3K9me2/me3 marks that are important for regulating the timeline of several processes during embryogenesis. These histone modifications are known to affect the timing of heterochromatin formation in early embryos [33,34]. H3K9me1/me2/me3 all contribute to the anchoring of heterochromatin to the nuclear periphery in both embryo and larval stages [33,35]. Loss of H3K9me2 specifically has been shown to result in delayed loss of developmental plasticity [36]. Deposition of the H3K9me2 modification in embryos is regulated by the SET domain-containing histone methyltransferase MET-2 [37,38].

Since *xol-1* is a zygotically transcribed gene that needs to be transcriptionally repressed for the proper development of a hermaphrodite embryo, it has been assumed to be a male-specific gene. It was shown that in early hermaphrodite embryos, *xol-1* is briefly transcribed at low levels [9] but it was assumed to be unlikely to be able to activate its downstream transcriptional program [9,39]. In this paper we confirm that *xol-1* is transcribed in hermaphrodite early embryos at low levels, but contrary to previous assumptions we also show that the loss of *xol-1* results in significant changes in the transcriptional profile of these early embryos. We also show that *xol-1* mutant embryos show accelerated hermaphrodite embryonic development as well as precocious establishment of dosage compensation and sex determination pathways. Finally, we show that several of these phenotypes are partially mediated through transcriptional regulation of the H3K9 histone methyltransferase MET-2.

## Results

### *xol-1* mutant hermaphrodites have accelerated embryonic development

To characterize the transcriptional changes in *xol-1* null mutants, we sequenced mRNA from synchronized *xol-1* hermaphrodite early embryos. We used the null mutant *xol-1 (ne4472)* that has previously been mapped and validated [40], and we observed an unexpectedly large number of transcriptional changes in the *xol-1* mutant compared to WT in early embryos (Fig 1A). The WT used in this study was the N2 Bristol strain [41]. We also performed RNA-seq on synchronized *xol-1* L1 larvae and observed that most of these transcriptional changes seen in early embryos are resolved at the L1 stage (Fig 1B). This high level of disruption of the regular transcriptional program in *xol-1* mutant embryos suggests that *xol-1* is expressed and is active in these early embryo stages. We confirmed this by analyzing datasets from Boeck et. al. (2016), where the authors generated a time-resolved transcriptome of *C. elegans* embryogenesis [42]. In this dataset, *xol-1* was detectable at the 4-cell stage, which was the earliest available time-point (Fig 1C). *xol-1* transcripts continued to accumulate in the embryos until 122min post fertilization (Fig 1C), which corresponds to early gastrulation. After this peak in expression, *xol-1* transcripts continue to decline but are still detectable in the late stages of embryogenesis when embryos obtain their 3-fold morphology at about 550 mins after fertilization (Fig 1C).

Genes that are known to disrupt embryonic developmental pathways generally tend to cause embryonic or larval lethality. In order to determine if disrupting *xol-1* in hermaphrodites results in any phenotypes, we performed embryonic and larval viability assays. Embryonic viability is a measure of the fraction of embryos that hatch into L1 larvae. Larval viability is a measure of the fraction of larvae that are able to develop into gravid adult worms. By both these measures, *xol-1* hermaphrodites were not significantly different from WT (Fig 1D), consistent with previous studies [10,40]. This is in contrast to male *xol-1* embryos, that have complete embryonic lethality [10]. The *xol-1* adult hermaphrodites also look, at least superficially, similar to healthy WT adults.

Since *xol-1* is important for turning on the male sex determination pathways, we hypothesized that it may be required in hermaphrodite embryonic development to potentiate the molecular pathways that lead to the development of the sperm producing structures in adult worms. To test this, we measured brood counts from *xol-1* mutant embryos and WT. In *C. elegans*, contractions in the oviducts force oocytes to pass through the spermatheca, where they get fertilized [43]. They emerge from the spermatheca as newly fertilized eggs that form a hard and impenetrable eggshell. After the spermatheca is emptied of sperm, oocytes continue to pass through without being fertilized and these oocytes do not form an eggshell. To quantify this process, we counted the number of fertilized and unfertilized eggs laid by worms from each genotype. The number of fertilized eggs were not significantly different between *xol-1* and WT, indicating that these worms did not have a defect in sperm production (Fig 1E). However, there was a small, but statistically significant difference in the total number of eggs laid, indicating a potential defect in oocyte production (Fig 1E).

Since the *xol-1* mutation in hermaphrodites did not seem to produce any obvious phenotypes, we performed gene ontology analysis on the *xol-1* early embryo RNA-seq dataset to determine what pathways are being affected in these mutants. Surprisingly, we found a recurrence of terms that were related to cell fate specification and cellular differentiation (Fig 1F). To determine the embryo stages represented in each sample, we DAPI-stained and imaged a subset of the population of synchronized *xol-1* and WT embryos. The worms were synchronized through repeated rounds of bleaching, and both samples were given the same amount of time to develop into young adults before they were bleached again to obtain early embryos.

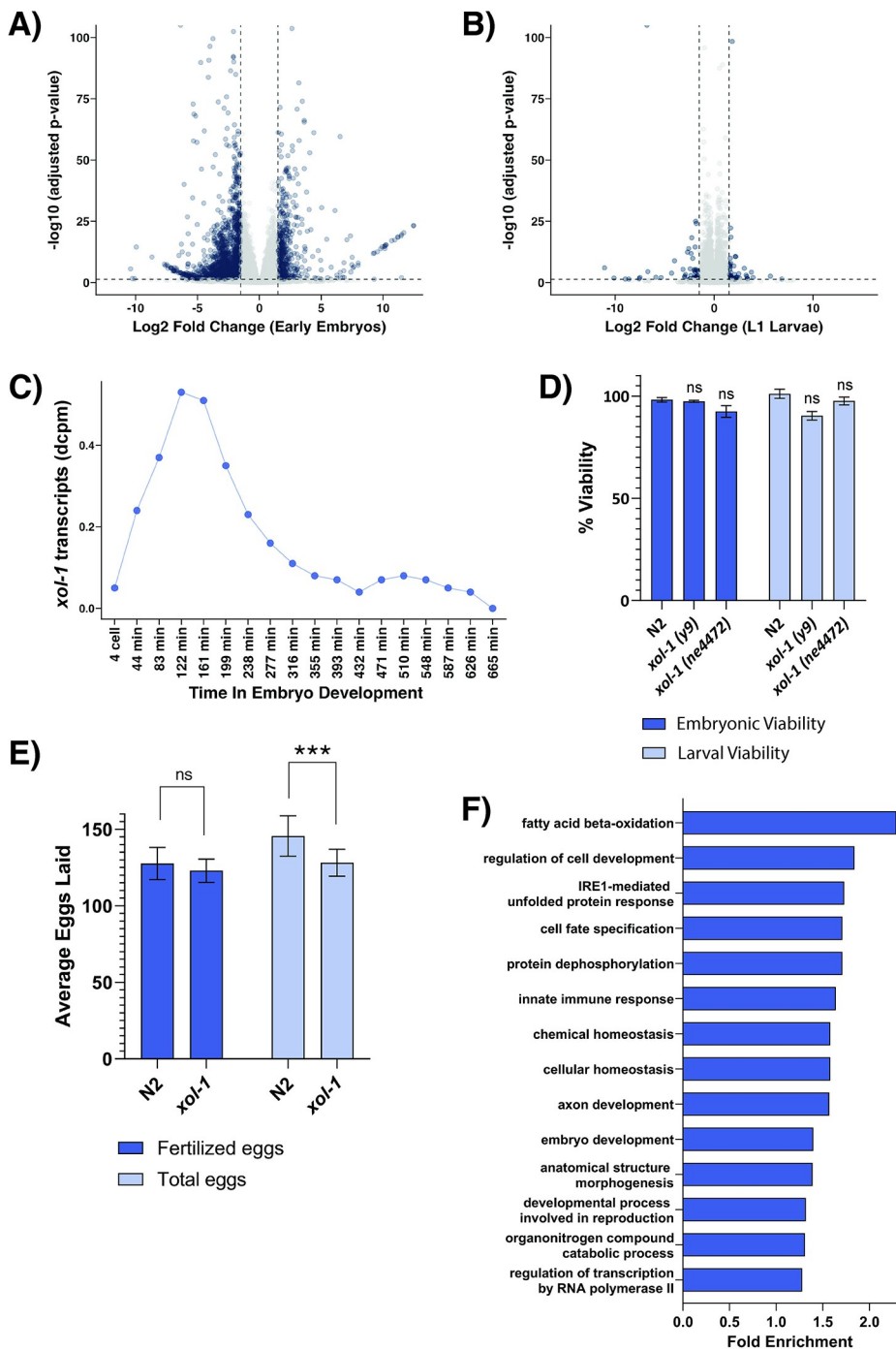

**Fig 1. Transcriptional changes and altered phenotypes associated with *xol-1* mutant early embryos.** (A) Volcano plot with genes differentially expressed in *xol-1* vs WT early embryo samples, absolute log2 fold change > 1.5 and adjusted p-value < 0.05. (B) Volcano plot with genes differentially expressed in *xol-1* vs WT L1 larvae, absolute log2 fold change > 1.5, adjusted p-value < 0.05. (C) Normalized transcript counts (depth of coverage per base per million reads (dcpm)) of *xol-1* gene during *C. elegans* embryogenesis. Dataset obtained from Boeck et. al. (2016) [42]. (D) Embryonic and larval viability scored in *xol-1* and WT. None of the differences are statistically significant. P-values obtained from chi-square test with null hypothesis assumption of no significant difference between populations. Error bars indicate SEM. N > 8. (E) Fertilized eggs laid and total eggs laid (fertilized and unfertilized) by *xol-1* and WT. Total eggs laid for *xol-1* vs WT, p-value = $9.9 \times 10^{-5}$. P-values obtained from chi-square test with null hypothesis assumption of no significant difference between populations. Error bars indicate SEM. n>12. (D-E) Comparisons that are not statistically significant are indicated by n.s, asterisks indicate level of statistical significance (* *p*<0.05; ** *p*<0.005; *** *p*<0.001). (F) Gene ontology analysis on *xol-1* vs WT early embryo RNA-seq.

Both WT and *xol-1* embryos were maintained at the standard 20˚C temperature. The histogram depicting the distribution of embryo stages in WT and *xol-1* (Fig 2A) shows that *xol-1* embryos are further along in embryonic development as measured by the number of cells in each embryo. Both *xol-1* and WT embryos have a peak abundance in 50–100 cell embryos, but *xol-1* mutants have a significantly higher proportion of late-stage embryos in the 100–200 cell and 200–300 cell categories. This leads us to two possibilities that could explain the difference in embryonic development: *xol-1* embryos could have an accelerated rate of cell division, or *xol-1* hermaphrodites could be laying embryos at a slower rate, which would result in the embryos present inside the worms to be more developed. Since we obtained our embryos by bleaching young adults, it is possible that the *xol-1* embryos released from the adults had had a longer time to develop.

In order to distinguish between these alternative hypotheses, we performed time-lapse experiments to directly observe embryonic development in real-time. There was no significant difference observed in the time taken to go from a 4-cell embryo to an 8-cell embryo (Fig 2B). However, there was a significant acceleration in the time taken in *xol-1* mutants to go from an 8-cell embryo to bean stage embryo (Fig 2C). In WT embryos, it takes roughly 4h 43m to reach bean stage from an 8-cell embryo, whereas in *xol-1* mutants the average time for this transition was 4h 29m. This difference is statistically significant at p-value < 0.05 using Welch's t-test. Later stages of embryonic development seem to be unaffected as the time taken to go from bean stage to 2-fold stage remained similar between the two genotypes (Fig 2D). This correlates well with the timeline of *xol-1* expression during embryogenesis (Fig 1C), where we see peak expression during early gastrulation and continued relatively high expression throughout gastrulation, corresponding to the 8-cell to bean stage. These results show that *xol-1* embryos follow an accelerated timeline of cell division.

To determine if this acceleration of embryonic development was restricted to higher rate of cell divisions without affecting gene expression programs that govern embryonic developmental pathways, or if the acceleration was more global, we analyzed whether the transcriptional state of our time-synchronized *xol-1* early embryo population resembled the transcription program from later-stage embryos. We used datasets from Spencer et. al. (2011), which quantified transcripts that were enriched at either early or late embryonic stages in WT [44]. Genes enriched in early embryos were significantly downregulated in the *xol-1* embryo population (Fig 2E). Conversely, we also found that genes enriched in late-stage embryos were significantly upregulated in the *xol-1* mutant embryos (Fig 2F). Genes enriched in L1 larvae populations were neither upregulated nor downregulated in *xol-1* embryos (S1A Fig). Together, these data suggest that a time-synchronized *xol-1* embryo population is developmentally accelerated compared to WT not only in terms of cell number and initiation of embryo morphogenesis, but also in the activation of late-stage embryonic transcriptional programs.

## Loss of *xol-1* leads to precocious loading of the dosage compensation complex

Since XOL-1 is known to have an essential role in the embryonic development of male *C. elegans* embryos [9,10], we looked at the known pathways in males that are regulated by XOL-1 to better understand its potential roles in hermaphrodite embryonic development. In males, XOL-1 represses the activity of the SDC proteins [9,10]. The SDC proteins independently regulate the pathways of sex determination and dosage compensation in hermaphrodites [17,45,46]. In male embryos where *xol-1* is highly expressed, male sex determination pathways are turned on and hermaphrodite sex determination and dosage compensation of the X chromosome are turned off [9,10] (Fig 3A). Based on this known mechanism of action, we

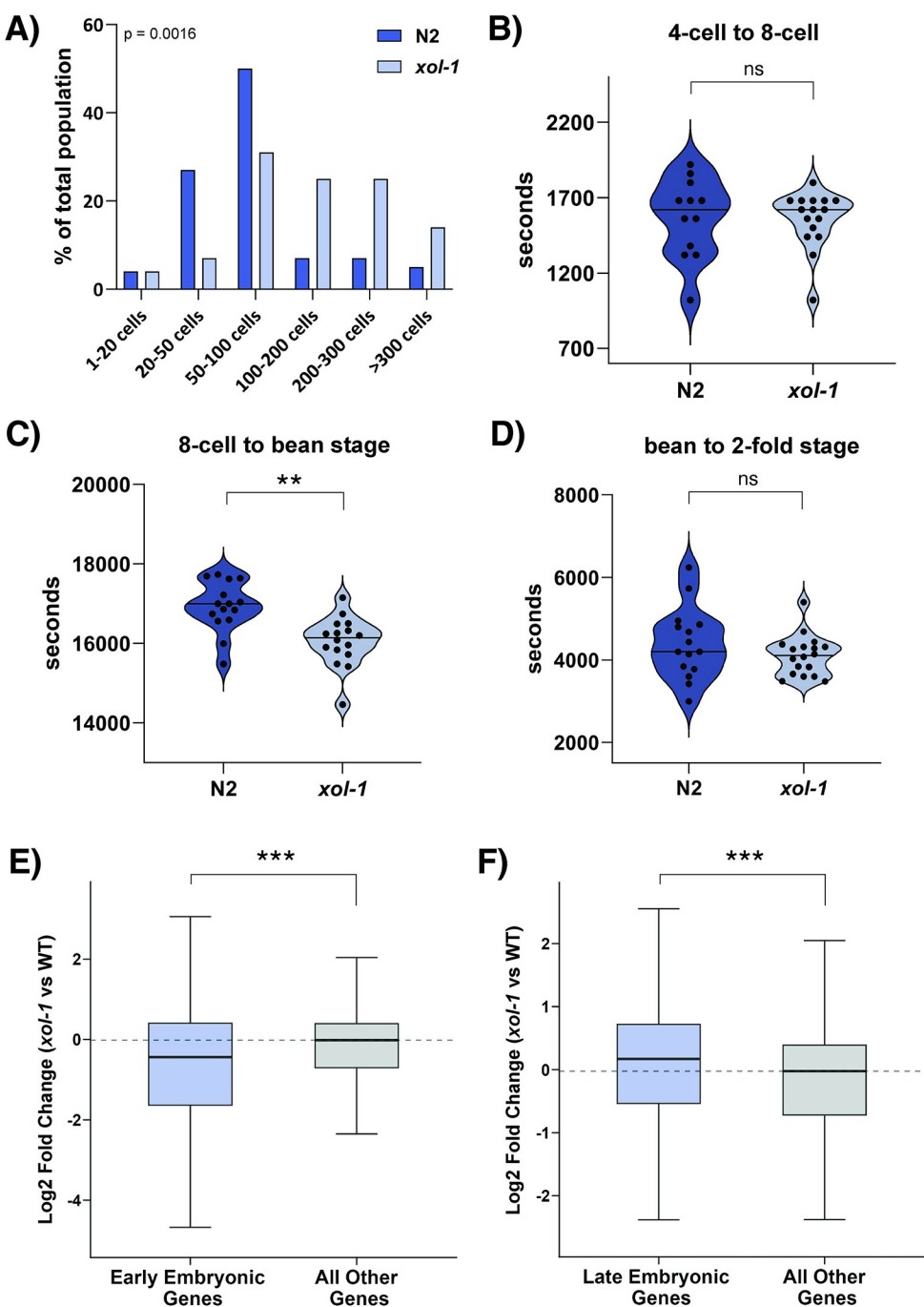

**Fig 2. *xol-1* embryos have an accelerated developmental timeline.** (A) Histogram depicting the distribution of embryos in *xol-1* and WT populations (n = 100). Embryos were binned into categories indicated on the x-axis based on number of nuclei visible by DAPI staining. Statistical test used was the chi-square test, p<0.05 (B-D) Time taken for *xol-1* and WT embryos to develop from (B) 4-cell embryo to 8-cell embryo (embryos scored: WT = 12, *xol-1* = 16) (C) 8-cell embryo to bean stage embryo (embryos scored: WT = 15, *xol-1* = 16) (p-value = 0.0004) and (D) bean stage embryo to 2-fold embryo. (embryos scored: WT = 15, *xol-1* = 18) P-values calculated using Welch's t-test with two-tailed distribution and unequal variance. (E) Boxplot depicting the median log2 fold change in *xol-1* vs WT for genes enriched in early embryos (left) and the rest of the dataset (right). p-value = 1.5 x 10$^{-6}$, wilcoxon rank-sum test. (F) Boxplot depicting the median log2 fold change in *xol-1* vs WT for gene enriched in late embryos (left) and the rest of the dataset (right). p-value = 3.6 x 10$^{-8}$, wilcoxon rank-sum test. Early and late embryo datasets obtained from Spencer et. al. (2011) [44]. Asterisks indicate level of statistical significance (* *p*<0.05; ** *p*<0.005; *** *p*<0.001, n.s not significant).

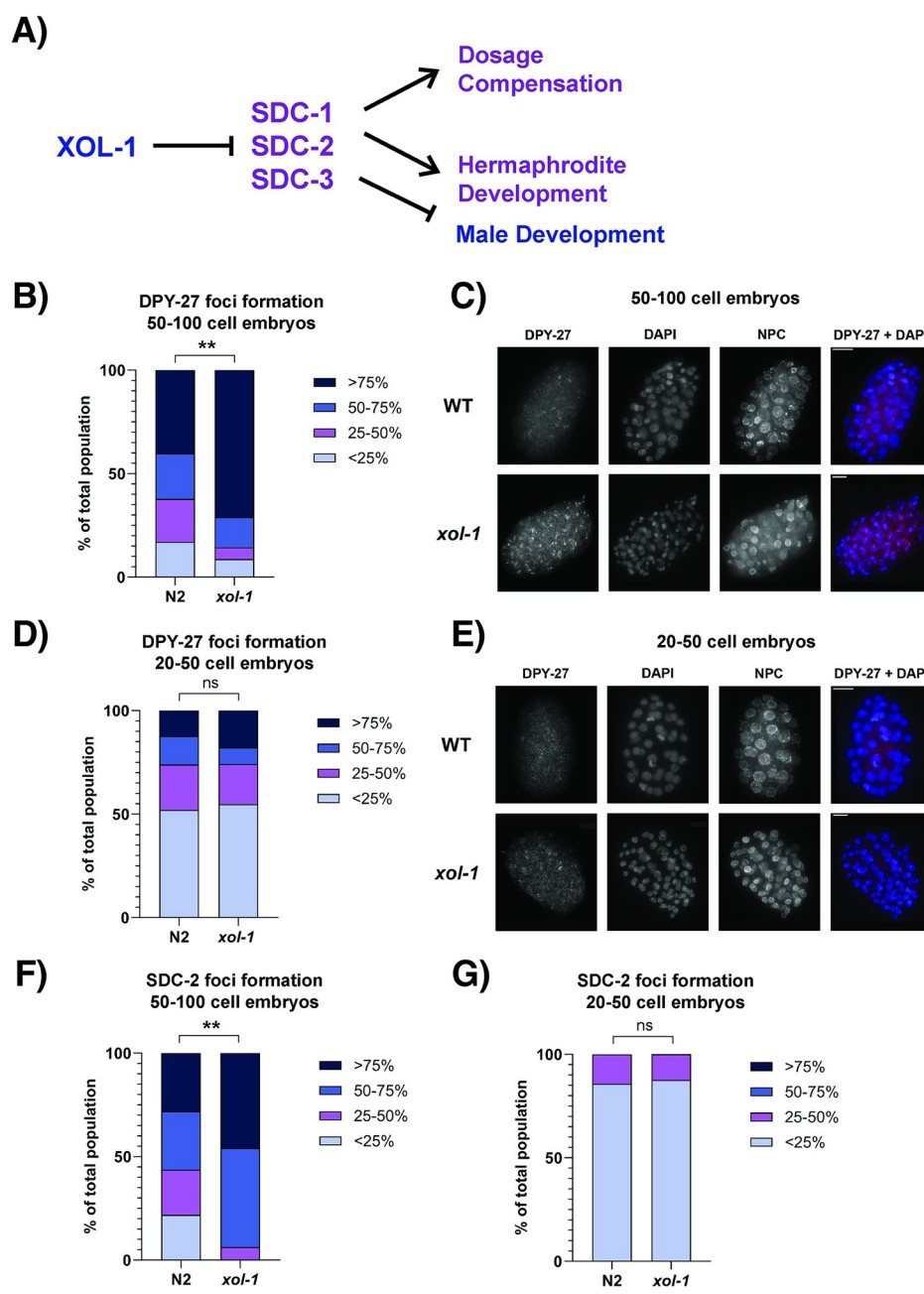

**Fig 3. Stage-matched *xol-1* embryos have precocious loading of the dosage compensation complex.** (A) Schematic depicting the known roles of *xol-1* in embryonic development. (B) Quantification of DPY-27 loading assay in 50–100 cell embryos. Embryos were scored and segregated into 4 categories based on the fraction of nuclei that had visible loading of DPY-27 on the X chromosomes in *xol-1* and WT. The categories were <25%, 25–50%, 50–75% and >75%. p = 0.0013. P-values obtained from chi-square test with null hypothesis assumption of no significant difference between populations. Staining against the nuclear pore complex (NPC) was used as an internal control. Embryos scored: N2 = 82, *xol-1* = 69. (C) Representative images for experiment quantified in (B). (D) Quantification of DPY-27 loading assay in 20–50 cell embryos. Embryos scored: N2 = 73, *xol-1* = 62 (E) Representative images for experiment quantified in (D). (F-G) Quantification of SDC-2 loading assay using *TY1::CeGFP::FLAG::sdc-2* and *TY1::CeGFP:: FLAG::sdc-2; xol-1* embryos in (F) 50–100 cell embryos (p = 0.002). Embryos scored: N2 = 32, *xol-1* = 48 (G) 20–50 cell embryos. Embryos scored: N2 = 14, *xol-1* = 16. Embryos scored with criteria similar to DPY-27 loading assay. Staining against the nuclear pore complex (NPC) was used as an internal control. Asterisks indicate level of statistical significance (* $p<0.05$; ** $p<0.005$; *** $p<0.001$, n.s not significant).

hypothesized that in *xol-1* hermaphrodite embryos, XOL-1 may be repressing dosage compensation pathways during the early stages of embryogenesis, thereby regulating the appropriate timing of the onset of dosage compensation in hermaphrodites.

To test this hypothesis, we performed immunofluorescence (IF) experiments in stage-matched *xol-1* mutant and WT embryos with antibodies against components of the dosage compensation complex (DCC) to determine when these components begin loading onto the X chromosomes during embryogenesis (Fig 3). The DCC consists of a 5 subunit condensin I$^{DC}$ complex and 5 additional proteins [17,22,23,45–50]. In Fig 3B–3E, we stained stage-matched 50–100 cell *xol-1* and WT embryos with antibodies against DPY-27, a core component of condensin I$^{DC}$. We also used a co-stain against the nuclear pore complex (NPC) as an internal control to confirm proper staining of embryos. We scored the embryos and binned them into 4 categories based on the percentage of cells inside an embryo that had visible localization of DPY-27 onto the X chromosomes. In WT embryos, DPY-27 begins loading onto the X chromosomes starting at the onset of gastrulation, when the embryo has about 26 cells [32,33,49]. At the 50–100 cell stages, DPY-27 localization on the X chromosomes in WT embryos remains stochastic (Fig 3B–3C, [32,33]). ~40% of embryos have DPY-27 localized to the X on >75% of nuclei in WT, and ~17% of WT embryos have DPY-27 localized to the X on <25% of nuclei (Fig 3B). Compared to this distribution, *xol-1* mutant embryos have a significantly higher proportion of embryos that have DPY-27 localized to the X. ~71% of *xol-1* embryos have DPY-27 localized to the X on >75% of nuclei, and only ~9% of *xol-1* embryos have DPY-27 localized to the X on <25% of nuclei (Fig 3B). At the earlier 20–50 cell stages, we do not see this precocious loading of DPY-27 in *xol-1* mutants (Fig 3D–3E).

We confirmed this early loading of the DCC by repeating the same experiment with SDC-2, another component of the DCC (Figs 3F–3G, S2A). SDC-2 is one of the additional proteins associated with the condensin I$^{DC}$ complex, and is required for the X-specific localization of all the other subunits of the DCC [51]. In 50–100 cell embryos, *xol-1* mutants have the same pattern of early loading of SDC-2 on the X chromosomes compared to WT (Fig 3F–3G). In *xol-1* embryos, ~46% of embryos have SDC-2 loading on the X in >75% of nuclei (Fig 3F). In contrast, only ~28% of WT embryos have SDC-2 loading in >75% of nuclei (Fig 3F). Similar to the DPY-27 loading experiment, we did not see a significant change in SDC-2 loading in 20–50 cell embryos (Fig 3G). The antibody concentrations in Fig 3B–3G were titrated to obtain the best dynamic range of foci staining between genotypes in the same experiment, and therefore they can be used to compare relative timing of foci formation between genotypes but is not a measure of absolute timing of loading for these DCC components. These experiments show that the loss of *xol-1* leads to the precocious loading of DCC components onto the X chromosome during hermaphrodite embryogenesis. This strengthens the hypothesis that *xol-1*, in addition to its role in turning off dosage compensation in male embryos, is also required in hermaphrodite embryos for regulating the timing of the initiation of dosage compensation.

## Loss of *xol-1* leads to changes in expression of sex determination pathways

Similar to the effect of *xol-1* on the dosage compensation pathway during embryogenesis, we hypothesized that the loss of *xol-1* would also affect sex determination pathways in the developing embryos. Since XOL-1 is responsible for repressing the hermaphrodite sex determination pathway in XO animals, we hypothesized that the *xol-1* mutant embryos would have upregulation of genes that are involved in hermaphrodite sex determination in XX early embryos as well. Conversely, since XOL-1 is responsible for activating the male sex determination pathways, we hypothesized that the loss of *xol-1* would result in the downregulation of genes involved in embryonic male sex determination.

To explore the differences in transcription between early male and hermaphrodite embryos, we generated male-enriched early embryo datasets using *him-8* mutants. The *him-8* gene is essential for the proper segregation of X chromosomes during meiosis [52,53]. In WT *C. elegans*, males occur at a very low frequency (~0.2%) [54]. Loss of *him-8* results in the overproduction of males due to the mis-segregation of the X chromosome, with an average of ~40% of the progeny being male worms [52]. This allowed us to generate samples with mixed-sex male and hermaphrodite embryos. The frequency of males in the population we collected for sequencing was ~45%. We performed successive rounds of synchronization using bleaching to obtain WT and *him-8* early embryo populations. We then sequenced mRNA from these samples, and used the differentially expressed genes to define gene sets that are enriched in the mixed-sex *him-8* embryos or the WT hermaphrodite embryos (Fig 4A). Genes that were upregulated in the *him*-8 embryo dataset were termed "male-biased" genes, and genes that were downregulated were termed "hermaphrodite-biased" genes.

We found that hermaphrodite-biased genes were upregulated (S3A Fig) and that male-biased genes were downregulated (S3B Fig) in *xol-1*/WT early embryo dataset. However, since *xol-1* embryos are developmentally more advanced compared to WT, it is possible that these differences are due to developmental stage rather than misregulation of male- and hermaphrodite-biased genes. To control for expression changes that might arise due to the mismatch in embryonic staging between the two genotypes, we used a conservative filtering strategy (Fig 4A). We used the time-resolved transcriptomic atlas of embryogenesis [42] to determine which genes increase and decrease with the progression of embryo development. Based on our working hypothesis, we would expect the hermaphrodite-biased genes to be significantly upregulated in *xol-1* early embryos. Therefore, we filtered out genes that increase in expression during embryogenesis from the hermaphrodite-biased genes (Fig 4B). Similarly for the male-biased genes, we filtered out genes that normally decrease in expression during embryogenesis (Fig 4B). This strategy eliminates the possibility that we would observe a false-positive enrichment of either gene set due to the accelerated development of *xol-1* embryos.

After filtering the gene sets obtained from *him-8*/WT RNA-seq, we observed that hermaphrodite-biased genes remained broadly upregulated in *xol-1* early embryos (Fig 4C). Male-biased genes were conversely downregulated in *xol-1* early embryos (Fig 4D). We also performed gene set enrichment analysis (GSEA) [55] on the *xol-1* early embryos and tested the enrichment of the two sex-biased gene sets (Figs 4E, S3C). GSEA is a computational technique that can quantify and statistically test the enrichment or depletion of any *a priori* defined set of genes, such as our sex-biased gene sets. Corroborating our previous analysis, GSEA showed that hermaphrodite-biased genes had a positive enrichment score of 3.16, while the male-biased genset had a negative enrichment score of -2.04 and both the gene sets were statistically significantly enriched and depleted respectively at p < 0.001 (Fig 4E). This data demonstrates that XOL-1 is responsible for promoting the expression of male-biased genes, and for repressing hermaphrodite-biased genes in early hermaphrodite embryos.

The disruption in hermaphrodite biased pathways appears to be more significant in terms of its effect size compared to the disruption in male biased pathways (Fig 4C–4E). This could be explained by the already very low transcription rate of male-specific genes in the WT early hermaphrodite embryos that the *xol-1* dataset is normalized against. The fact that *xol-1* L1 larvae do not continue to have a broadly disrupted transcriptional profile (Fig 1B) indicates that there are compensatory mechanisms that may be activated later in embryonic development.

XOL-1 represses the activity of the SDC proteins [9,10], that are known to regulate the hermaphrodite and male sex determination pathways [17,45,46]. Though the mechanism of this regulation has not yet been elucidated, it has been hypothesized that XOL-1 may transcriptionally regulate the *sdc* genes [56]. We looked at the expression of the *sdc* genes *sdc-1*, *sdc-2* and

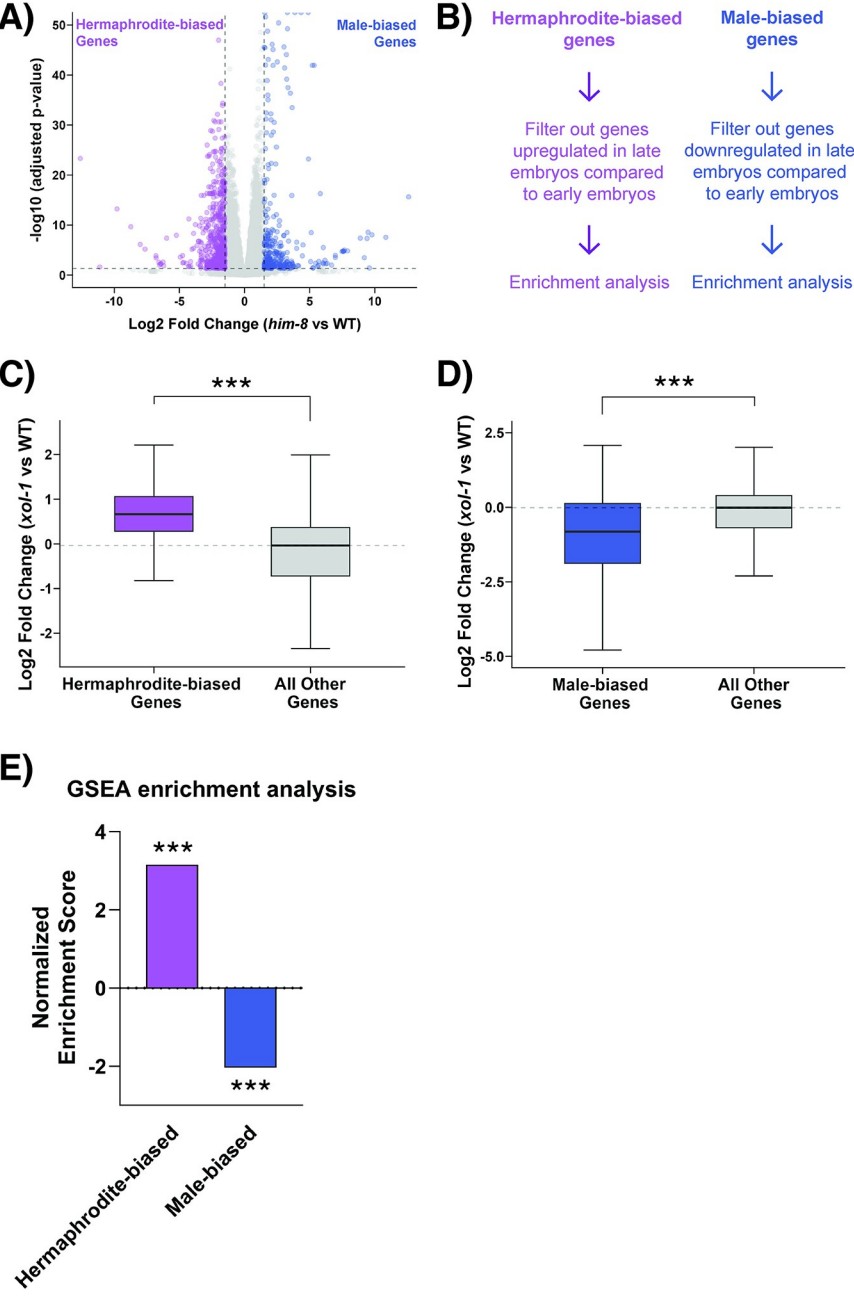

**Fig 4. XOL-1 regulates sex determination pathways in hermaphrodite early embryos.** (A) Volcano plot with genes differentially expressed in *him-8* vs WT early embryo samples, absolute log2 fold change>1.5 and adjusted p-value<0.05. Significantly upregulated genes were used to generate the "male-biased" gene set and downregulated genes were used to generate the "hermaphrodite-biased" gene set (B) Schematic representing the filtering strategy used to refine sex-biased gene sets. Time-resolved dataset from Boeck et. al. (2016) [42] was used to filter gene sets. (C-D) Boxplot showing median log2 fold change in *xol-1* vs WT for (C) hermaphrodite-biased gene set (left) (p < 2.2 x 10$^{-16}$) or (D) male-biased gene set (left) (p = 0.01) and the rest of the dataset (right). p-values obtained from wilcoxon rank-sum test (E) Gene set enrichment analysis on *xol-1* vs WT dataset using sex-biased gene sets. Asterisks indicate level of statistical significance (* *p*<0.05, ** *p*<0.005, *** *p*<0.001, n.s not significant).

*sdc-3* in *xol-1* mutant embryos to test this hypothesis. We did not observe any transcriptional upregulation of the *sdc* genes in our dataset (S3D Fig), suggesting that XOL-1 does not directly regulate the transcription of these genes, at least in hermaphrodites. In addition, we looked at our *him-8* RNA-seq dataset to determine if the *sdc* genes are repressed in males. We did not observe a significant downregulation of the *sdc* genes in *him-8* mutants (S3E Fig). In both these cases, we saw a transcriptional change in the opposing direction i.e. *sdc-2* was significantly lower in *xol-1* mutants (S3D Fig), and all three *sdc* genes were significantly higher in *him-8* mutants (S3E Fig). This suggests that in both males and hermaphrodites, XOL-1 is not transcriptionally repressing the *sdc* genes to regulate their activity. Furthermore, we quantified the intensity of SDC-2 protein signal in 50–100 cell embryos in WT and *xol-1* mutants to determine whether SDC-2 proteins levels are higher in a *xol-1* mutant background (S2C Fig). Similar to the mRNA levels, we observed a decrease, rather than the expected increase in SDC-2 protein levels (S2C Fig). These results indicate that XOL-1 is not repressing *sdc-2* either transcriptionally or translationally, and that the observed precocious loading of SDC-2 protein to the X observed in *xol-1* mutants (Fig 2F) is not due to increased protein levels.

### MET-2 and ARLE-14 are differentially expressed in *xol-1* mutants

In order to further probe the mechanism behind these disruptions we took a closer look at the specific genes that were misregulated in *xol-1* mutant early embryos. Among the genes that were significantly differentially expressed, some of the most interesting were regulators of H3K9 methylation. These regulators are responsible for the deposition of repressive histone modifications on the chromatin during embryogenesis [34,36,37], and several components of this pathway are involved in dosage compensation [25]. An altered H3K9 chromatin landscape during embryogenesis is linked to disruption of the timeline of heterochromatin formation, and the loss of developmental plasticity [34,36]. Simultaneously, the cellular differentiation programs in the embryos can also be altered by disrupting the regulators of this pathway [35,36]. One of these regulators is MET-2, a SET-domain containing histone methyltransferase that is responsible for the deposition of H3K9me1/me2 modifications on chromatin [37]. Loss of *met-2* disrupts dosage compensation in adult *C. elegans* [25], as measured by the small but statistically significant de-compaction of the X chromosome. *met-2* was significantly upregulated in *xol-1* early embryos (Fig 5A). *arle-14*, a binding partner of MET-2 that is known to strengthen the association of MET-2 with chromatin [38], was also found to be significantly upregulated (Fig 5A). Since the *xol-1* mutant embryos are time-synchronized with WT but not synchronized in terms of their embryonic development, we also looked at time-resolved embryogenesis datasets [42] to determine whether the change in *met-2* and *arle-14* expression could be explained by the difference in embryonic development between *xol-1* and WT embryos. We found that both *met-2* and *arle-14* expression decreases with embryonic age in WT (Fig 5B). This is in stark contrast to the pattern of upregulation seen in *xol-1* early embryos (Fig 5A). We also looked at *met-2* expression in late-embryo samples, and did not see a statistically significant difference in expression between *xol-1* and WT (S4A Fig). This suggests that *met-2* may be a direct or indirect downstream target of the XOL-1 pathway.

Several other regulators of H3K9 methylation are also differentially expressed. These include *set-32*, which contributes to germline deposition of H3K9me3 [57], and which was significantly upregulated in *xol-1* mutants (Fig 5C). *set-25*, which is responsible for the deposition of H3K9me3 [37], was also significantly upregulated (Fig 5C). Some additional accessory proteins involved in H3K9 methylation or recognition include *lin-61*, *nrde-3* and *cec-4* [34,35,58,59], which were all significantly downregulated in *xol-1* early embryos (Fig 5C). However, the expression of these three genes is known to decrease during the progression of

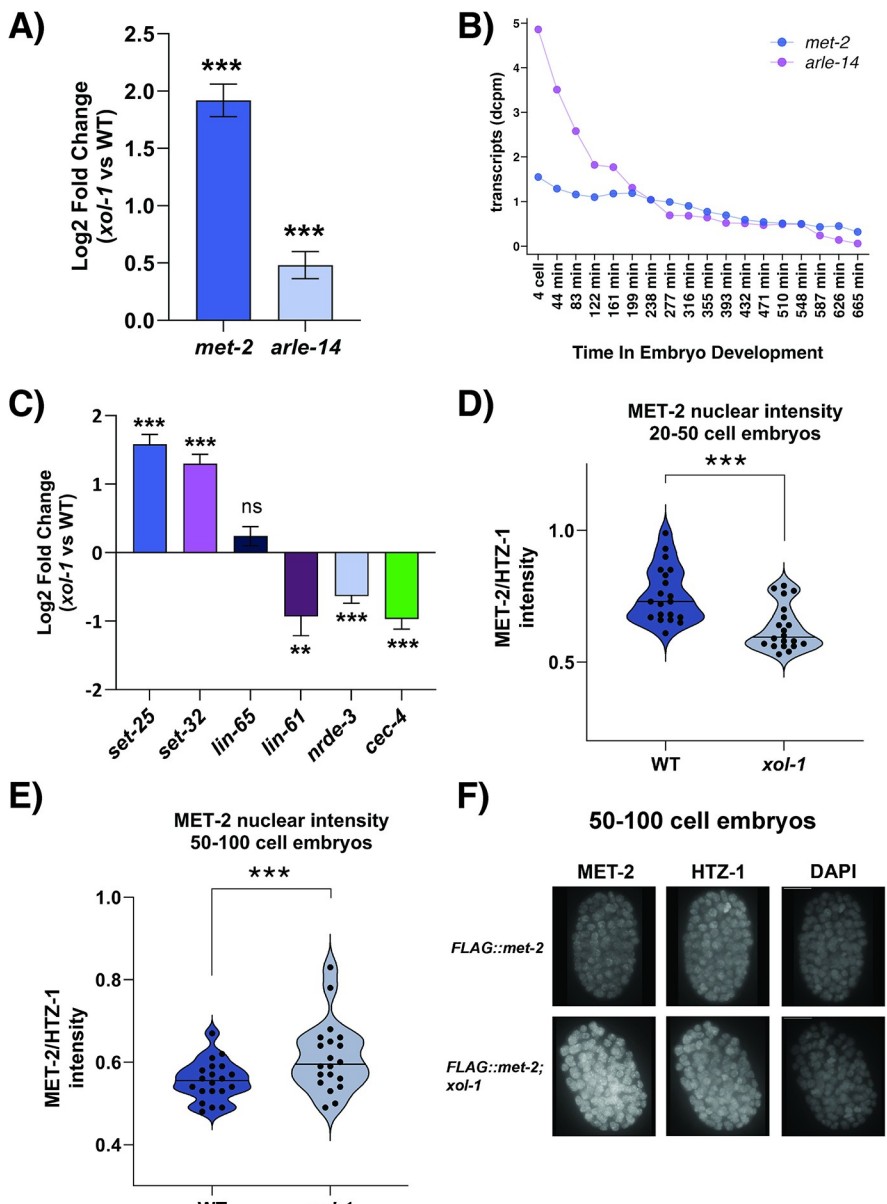

**Fig 5. MET-2 and its co-factor ARLE-14 are upregulated in *xol-1* embryos.** (A) Transcript levels (dcpm) of *met-2* and *arle-14* in *xol-1* vs WT RNA-seq. Adjusted p-value is determined by wald test with Benjamini-Hochberg correction using DESeq2. Error bars indicate standard error IfcSE. (B) Normalized transcript counts of *met-2* and *arle-14* genes during *C. elegans* embryogenesis. Dataset obtained from Boeck et. al. (2016) [42] (C) Transcript levels of *lin-65*, *lin-61*, *nrde-3* and *cec-4* in *xol-1* vs WT RNA-seq. Error bars indicate standard error IfcSE (D) 3xFLAG::MET-2 protein levels in *xol-1* and WT 20–50 cell embryos. p = 0.0001. MET-2 intensity was normalized to HTZ-1 co-stain. P-values were calculated using Welch's t-test with two-tailed distribution and unequal variance. 20 nuclei were quantified in total from 7 distinct embryos for each genotype. (E) 3xFLAG::MET-2 protein levels in *xol-1* and WT 50–100 cell embryos. p = 0.01. 20 nuclei were quantified in total from 10 distinct embryos for each genotype. Error bars indicate SEM. (F) Representative images of stained embryos quantified in (E). Asterisks indicate level of statistical significance (* *p*<0.05, ** *p*<0.005, *** *p*<0.001).

embryogenesis (S4B Fig). Therefore, we cannot accurately determine if this downregulation is due to the mismatch in embryonic age between the two genotypes or due to regulation by XOL-1.

Since *met-2* transcripts are significantly upregulated in *xol-1* embryos, and H3K9 methylation is known to regulate the timeline of important developmental milestones during embryogenesis [34,36], we sought to confirm that nuclear accumulation of MET-2 protein is also increased in *xol-1* mutants. MET-2 is known to be localized in the cytosol in early embryos and it is transported inside the nucleus starting at 20–50 cell embryos by its co-factor LIN-65 [34,58]. Nuclear accumulation of MET-2 reaches its peak level at the 50–100 cell stage [34]. We measured 3XFLAG::MET-2 nuclear protein levels in staged-matched *xol-1* mutant and WT embryos using immunofluorescence (IF) imaging with anti-FLAG antibodies. At the 20–50 cell stage, MET-2 was statistically significantly lower in *xol-1* compared to WT (Fig 5D). However, there was a significant increase in nuclear MET-2 levels at the 50–100 cell stage (Fig 5E–5F). LIN-65, a co-factor of MET-2, regulates the transport of MET-2 into the nucleus and is known to be the rate-limiting step in the nuclear accumulation of MET-2 [34]. Since *lin-65* expression is not significantly altered in *xol-1* embryos (Fig 5C), MET-2 is likely transported into the nucleus by LIN-65-mediated mechanism at the same rate in both WT and *xol-1* embryos. Progressive accumulation of MET-2 protein by LIN-65 is only visible at the 50–100 cell stages, where peak nuclear MET-2 protein levels are higher in *xol-1* embryos (Fig 5E–5F).

## *xol-1* mutants have altered levels of H3K9me2/me3

MET-2 is responsible for the *de novo* deposition of repressive H3K9me1/me2 on chromatin after fertilization [37]. Inducing premature nuclear accumulation of MET-2 using a nuclear localization signal (NLS) leads to a premature increase in H3K9me2 on the chromatin [34]. Since nuclear MET-2 protein levels are increased in *xol-1* mutant embryos, the next pertinent question is whether the H3K9 methylation status of the chromatin in these mutants is similarly altered. To test this, we used IF to measure the intensity of H3K9me2 and H3K9me3 on chromatin in stage-matched *xol-1* embryos (Fig 6A–6B). H3K9me2 was normalized against pan-H3, and H3K9me3 was normalized against HTZ-1 (See Methods). Contrary to our expectations, we found that H3K9me2 was significantly decreased in *xol-1* mutants compared to WT in both 20–50 cell (Fig 6A) and 50–100 cell embryos (Fig 6B). At the same time, we found that H3K9me3 intensity measured through IF was significantly increased in *xol-1* mutants at both the 20–50 cell and 50–100 cell stages (Fig 6C–6E). The marked increase in H3K9me3 in *xol-1* mutants suggests that MET-2-mediated H3K9me2 is being used as a substrate by SET-25 for the accumulating deposition of H3K9me3. Taken together, this data confirms that mis-expression of *met-2* and other components of the H3K9 pathway are contributing to an altered H3 landscape in *xol-1* embryos.

## Loss of *met-2* in *xol-1* mutant background leads to the reversal of accelerated development

To test if the embryonic phenotypes in *xol-1* are mediated by the upregulation of *met-2*, we generated a *met-2; xol-1* double mutant strain. We first sought to test if introducing a *met-2* mutation would abrogate the precocious loading of the DCC onto the X chromosome in *xol-1* mutants observed in Fig 3. We compared *xol-1*, *met-2* and *met-2; xol-1* mutants in this assay, and scored 50–100 cell embryos based on what fraction of nuclei in each embryo had visible loading of the DCC on the X chromosomes (Fig 7A). We also compared the rate of DPY-27 foci formation between WT and *met-2* mutants (S4C–S4D Fig), and found that foci formation occurs at a similar rate in both genotypes. *xol-1* and *met-2* mutants were significantly different in their distributions, with *xol-1* having its largest fraction of embryos in the >75% category and *met-2* having its largest fraction in the <25% category. The *met-2; xol-1* double mutant seemed to display a distribution that is somewhere in between the two single mutants. *met-2;*

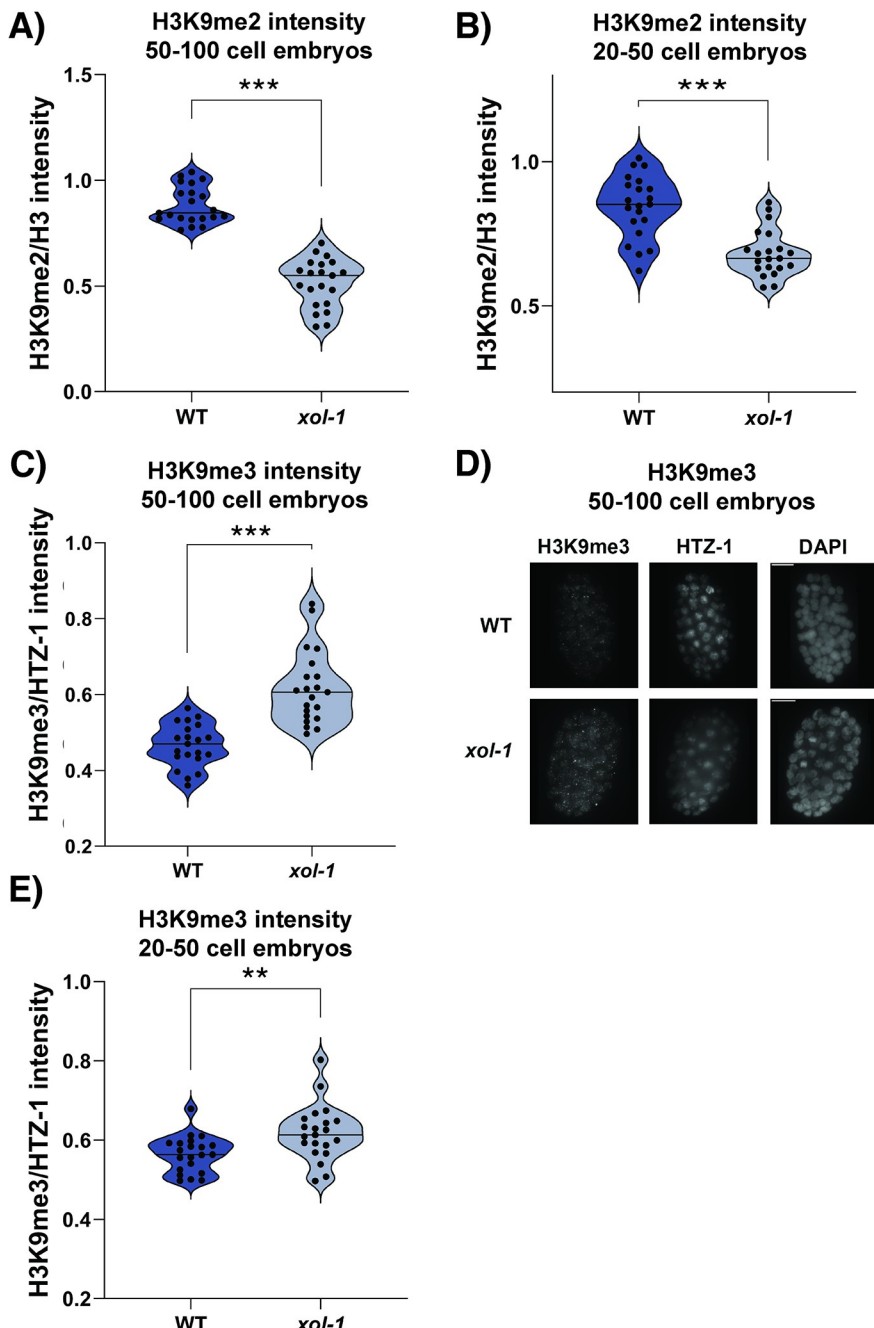

**Fig 6. H3K9 methylation is altered in stage-matched *xol-1* embryos. (A-C, E)** Intensity quantification of staged *xol-1* and WT embryos. **(A)** H3K9me2/H3 in 20–50 cell embryos, p = 2.9 x 10$^{-6}$ and **(B)** H3K9me2/H3 in 50–100 cell embryos, p = 4.2 x 10$^{-14}$. **(C)** H3K9me3/HTZ-1 in 50–100 cell embryos, p = 6.1 x 10$^{-7}$. **(D)** Representative images of staining quantified in (C) and **(E)** H3K9me3/HTZ-1 in 20–50 cell embryos, p = 0.005. P-values were calculated using Welch's t-test with two-tailed distribution and unequal variance. 21 nuclei were quantified in total for each condition, representing nuclei from at least 7 embryos. Asterisks indicate level of statistical significance (* *p*<0.05, ** *p*<0.005, *** *p*<0.001, n.s not significant). Error bars indicate SEM.

*xol-1* had markedly fewer embryos in the >75% category compared to *xol-1*, and was statistically significantly different from both single mutants. These results suggest that some, but not all, of the accelerated loading of the DCC onto the X seen in *xol-1* may be attributed to the

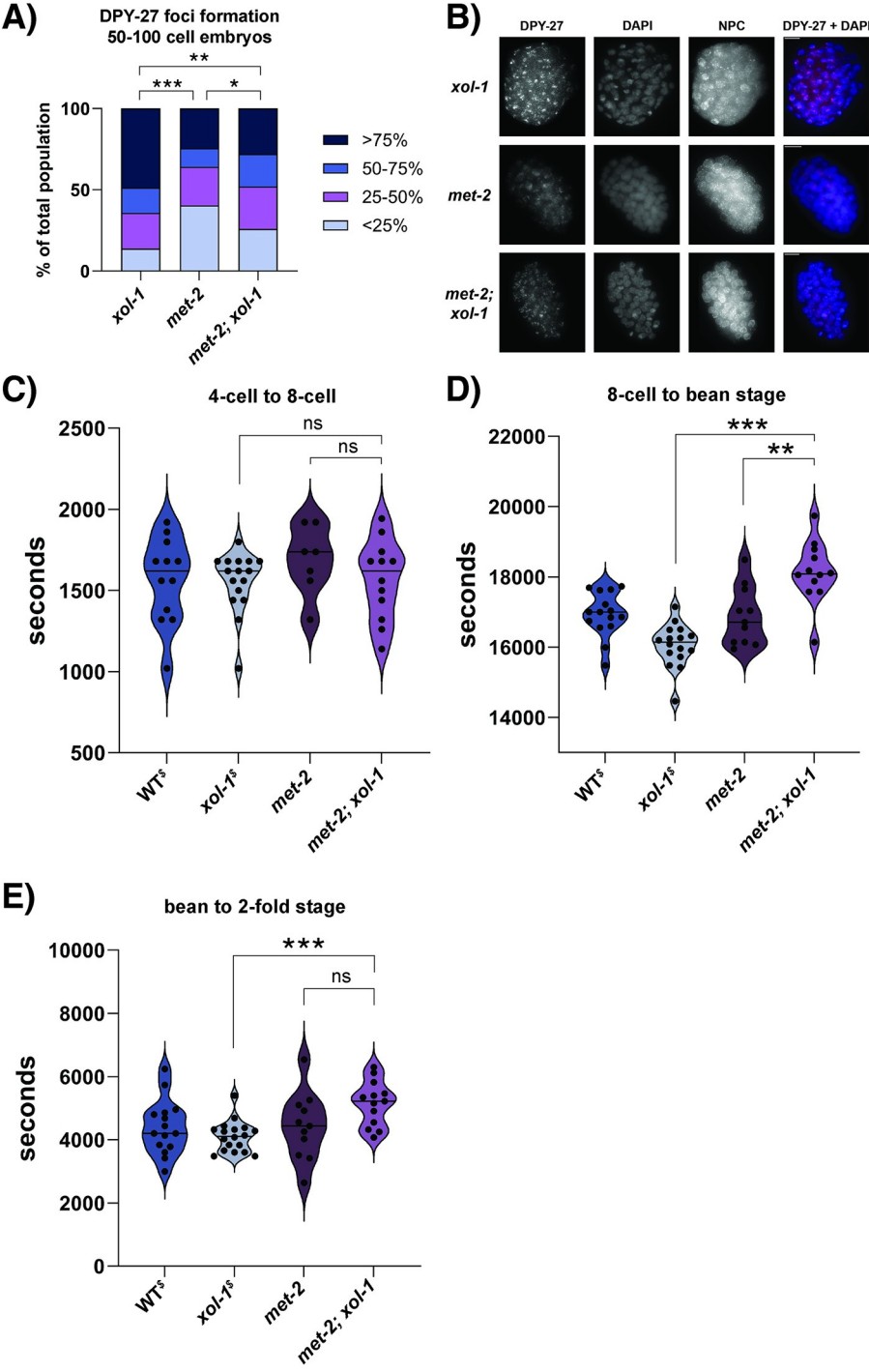

**Fig 7. Loss of *met-2* in *xol-1* background leads to the reversal of some *xol-1* phenotypes.** (A) Quantification of DPY-27 loading assay in *xol-1*, *met-2* and *met-2; xol-1* 50–100 cell embryos. Embryos were scored and segregated into 4 categories based on the fraction of nuclei that had visible loading of DPY-27 on the X chromosomes. The categories were <25%, 25–50%, 50–75% and >75%. *xol-1* vs *met-2*, p < 0.00001, *met-2* vs *met-2; xol-1*, p = 0.041, *xol-1* vs *met-2; xol-1*, p = 0.003. P-values obtained from chi-square test with null hypothesis assumption of no significant difference between populations. Staining against the nuclear pore complex (NPC) was used as an internal control. Embryos scored: *xol-1* = 123, *met-2* = 139, *met-2; xol-1* = 150 (B) Representative images for experiment quantified in (A). (C-E) Time taken for WT, *xol-1*, *met-2* and *met-2; xol-1* embryos to go from (C) 4-cell embryo to 8-cell embryo (embryos scored: N2 = 12, *xol-1* = 16, *met-2* = 7, *met-2; xol-1* = 12) (D) 8-cell embryo to bean stage embryo, *xol-1* vs *met-2; xol-1*, p = 1.16 x 10$^{-6}$, *met-2* vs *met-2; xol-1*, p = 0.001 (embryos scored: N2 = 15, *xol-1* = 16, *met-2* = 11, *met-2; xol-1* = 12)

and (E) bean stage embryo to 2-fold embryo, *xol-1* vs *met-2; xol-1*, p = 0.0001 (embryos scored: N2 = 15, *xol-1* = 18, *met-2* = 11, *met-2; xol-1* = 13). (C-E) P-values calculated using Welch's t-test with two-tailed distribution and unequal variance. Asterisks indicate level of statistical significance (* $p<0.05$; ** $p<0.005$; *** $p<0.001$, n.s not significant). $^{\$}$ notation on WT$^{\$}$ and xol-1$^{\$}$ indicates that the data is from the same experiment as Fig 2B–2D.

upregulation of *met-2*. There are likely additional players involved in this process as in the *met-2; xol-1* double mutant this phenotype was only partially reversed.

The most striking phenotype in *xol-1* early embryos is the acceleration of embryonic development (Fig 2B–2D). We performed time-lapse experiments on the *met-2; xol-1* mutant to test if the development of embryos is still accelerated when *met-2* is lost in a *xol-1* background. We compared the time taken for the embryos to go from 4-cell to 8-cell (Fig 7C), 8-cell to bean stage (Fig 7D) and from bean stage to 2-fold stage (Fig 7E). We also quantified the timeline of development of *met-2* embryos as a control and compared both these genotypes to the WT and *xol-1* embryos tested in Fig 2. *met-2* mutants are known to have defects in embryonic viability [36,38], and we empirically observed that *met-2; xol-1* mutants also seemed to have some embryonic lethality. For this analysis, we eliminated any embryos that were not able to reach 2-fold stage. *met-2* mutants were not significantly different from WT in terms of the timeline of embryo development at any of the time-points tested (Fig 7C–7E). None of the genotypes were significantly different in the time taken to go from 4-cell to 8-cell (Fig 7C). Compared to *xol-1* mutants, *met-2; xol-1* double mutants were significantly delayed in their transition from 8-cell to bean stage (Fig 7D), as well as bean stage to 2-fold stage (Fig 7E). These results suggest that loss of *met-2* completely abrogates accelerated development caused by loss of *xol-1* in hermaphrodite embryos.

Taken together, these results suggest that *xol-1* has an important role in regulating the timeline of embryonic development in hermaphrodites. These roles include regulating the speed of embryonic development during the early phases of embryogenesis, as well as the timing of initiation of sex-specific transcriptional programs. The disruption of *xol-1* results in an acceleration of embryonic development and a measurable disruption of developmental pathways.

## Discussion

The data presented in this paper suggests that *xol-1* has an important role as a developmental regulator of embryogenesis in hermaphrodites. Loss of *xol-1* leads to accelerated development of hermaphrodite embryos, precocious initiation of dosage compensation as well as broad misregulation of sex determination pathways. We also present data that suggests that *met-2*, an H3K9 methyltransferase, is a downstream target of *xol-1* that regulates several of these pathways.

### The mechanism of XOL-1-mediated repression of the SDC proteins

Though the *xol-1* gene has very little nucleotide and protein sequence homology with any non-nematode genes, the structure of XOL-1 resolved by x-ray crystallography revealed that it has significant structural homology to GHMP kinase family members [39]. However, ATP binding assays failed to detect any ATP binding by XOL-1 under standard conditions *in vitro* [39]. The authors of this study speculated that with some conformational changes XOL-1 may be able to bind polynucleotide substrates, due to sub-domain specific homology to RNA/DNA-binding proteins [39]. Beyond these putative mechanistic insights derived from this structural study of XOL-1, there is not much evidence that sheds light on how XOL-1 may be repressing the activity of the SDC proteins.

A review by Meyer et al. (2004) speculated that XOL-1 may be similar in its mechanism to GAL3p, a GHMP kinase family member in *S. cerevisiae*, that is known to act as a transcriptional inducer of GAL genes [56]. However, *sdc-1*, *sdc-2* and *sdc-3* genes are not significantly upregulated in *xol-1* embryos (S3D Fig), or significantly downregulated in *him-8* embryos (S3E Fig), suggesting that XOL-1 doesn't transcriptionally regulate the expression of the *sdc* genes. Furthermore, SDC-2 protein levels were also not significantly higher in *xol-1* embryos (S2C Fig), suggesting that XOL-1 also likely does not regulate mRNA translation or SDC-2 protein stability. This suggests several possibilities for the mechanism of action of XOL-1. One of these is that XOL-1 is regulating the activity of the SDC proteins at the level of protein-protein interactions or post-translational modifications, which is compatible with its current designation as a putative GHMP kinase. This first possibility is challenged by the lack of ATP binding by XOL-1 under standard conditions. A second possibility arises from the assumption that XOL-1 may able to bind DNA sequences after a conformational change and that it may be able to directly regulate transcription of genes. In this case, it is likely that there are additional regulators downstream of XOL-1, but upstream of the *sdc* genes. Since no studies have identified regulators of SDC activity beyond XOL-1, the existence of these as yet unknown mediators is plausible. Luz et al. (2003) noted particular structural similarity between RNA-binding protein domains and C-terminal domains of GHMP kinases such as XOL-1 [39], raising another possible mechanism of XOL-1 action through the modulation of mRNA translation following RNA-binding.

## Regulation of developmental timing by XOL-1

We observed the striking phenotype of accelerated embryogenesis in *xol-1* mutant embryos. There are reports of accelerated development independent of temperature changes in other organisms. In these cases the acceleration is often due to altered hormonal control of embryonic growth rate [60–64]. However, to our knowledge there are no reported accounts of accelerated embryonic development that are independent of temperate shift in *C. elegans*. Instead, perturbations usually lead to delayed embryonic development [65–67]. There are no known hormonal signaling pathways that have been shown to be important in regulating *C. elegans* embryogenesis, though some hormones are involved in regulating developmental timing at larval stages [68]. Furthermore, in *xol-1* mutants we saw acceleration in two different ways. First, we observed that cell division was faster between the 8-cell and bean stages of embryonic development. Concurrently, we observed that these embryos had a more advanced late-embryonic transcriptional program which matched with their accelerated cell division. The second mode of acceleration was the observed precocious induction of dosage compensation on the X chromosomes. Here, stage-matched 50–100 cell embryos were further along in *xol-1* mutants in their initiation of dosage compensation.

*xol-1* appears to be a *Caenorhabditis*-specific gene, with no known homologs in any other genus. *C. briggsae*, another nematode species that diverged from *C. elegans* roughly 100 million years ago [69], has an orthologous *xol-1* gene [39]. Even between these two closely related species, the protein sequence similarity was found to be low, though there was significant conservation in the protein domains believed to be important for its putative GHMP kinase activity [39]. In addition to the validated *xol-1* ortholog in *C. briggsae*, WormBase lists putative orthologs in several other *Caenorhabditis* species (https://wormbase.org/species/c_elegans/gene/WBGene00006962#0-9fc1bdg6-10, last accessed March 14, 2024). This suggests that XOL-1 is a recently evolved gene within the *Caenorhabditis* genus, where it has assumed the role of the master regulator of sex determination, dosage compensation and as our data shows, a novel regulator that is responsible for controlling the timing of embryonic development.

## Robustness in developmental regulation of embryogenesis

Despite the significant difference in the timeline of cell division and broad disruption in the timeline of initiation of dosage compensation and sex determination pathways, the *xol-1* embryos show no significant embryonic or larval lethality (Fig 1D). In addition, we see that most of the significant transcriptional differences in *xol-1* vs WT in early embryos are resolved by the time the worms reach L1 larval stage (Fig 1A, 1B). The upregulation of *met-2* in *xol-1* embryos (Fig 5A) is also resolved in late embryos (S4A Fig). These data suggest that there may be redundant or compensatory mechanisms that regulate these disrupted pathways during late embryogenesis and larval stages. This robustness in embryonic development has been described before in other organisms, where quite significant disruptions in early embryogenesis can be compensated for at later stages resulting in perfectly viable larvae and adults [70,71]. Even in *C. elegans*, complete loss of heterochromatin formation in *set-25* early embryos does not result in significant embryonic lethality [34]. In another related pathway, *cec-4* early embryos with complete loss of heterochromatin tethering also have completely viable embryos [35]. In the case of *cec-4*, compensatory pathways are activated in the larval stages that regulate the tethering of heterochromatin to the nuclear periphery mediated by MRG-1 [72]. Similar compensatory pathways could be involved in resolving disruptions caused by misregulation of sex determination pathways in *xol-1* mutants.

## MET-2 acts as a downstream effector of XOL-1

The data in this paper also demonstrates that mutations in *met-2* are able to reverse some of the phenotypes we have observed in *xol-1* mutant embryos, including the precocious loading of the DCC complex onto the X chromosomes (Fig 3A and 3E), and the acceleration of the timeline of embryonic cell divisions (Fig 2A and 2C). This data suggests that *met-2* is one of the downstream effectors of *xol-1*. Since there is a distinct lack of any mechanistic insight into how XOL-1 may be regulating any of its downstream targets, including the SDC proteins, it is also difficult to determine whether *met-2* is regulated directly or indirectly by XOL-1. In the case of the timeline of embryogenesis, *met-2; xol-1* embryos appeared significantly delayed compared to *xol-1*, but also compared to WT and *met-2* embryos. This suggests that *met-2* may play a broader role in the regulation of the timeline of embryogenesis. This is supported by the available evidence on the role of *met-2* during embryogenesis. MET-2-mediated deposition of H3K9me2, and SET-25-mediated deposition of H3K9me3 is important for regulating the timing of heterochromatin formation in *C. elegans* embryos [34]. In addition, *met-2*, *set-25* and *met-2; set-25* embryos all show significant disruptions in the timeline of loss of developmental plasticity and acquisition of differentiated cellular identity [36].

In the case of precocious loading of the DCC, *met-2; xol-1* embryos have partial reversal of the phenotype compared to *xol-1* (Fig 7A). The current known role of *met-2* in dosage compensation is that of an accessory pathway that reinforces X repression [25]. *met-2* deposits the H3K9me2 repressive mark, which is mainly found on the left arms of the X chromosomes of hermaphrodites [73]. There isn't any evidence for a direct interaction between the components of the DCC and *met-2*, or between the DCC and any other regulators of H3K9 methylation. As mentioned earlier, *met-2* has been shown to alter the timing of loss of developmental plasticity during embryogenesis [36]. In addition, the onset of dosage compensation in embryos is linked to the timing of loss of developmental plasticity [32]. Given this, it is likely that the delay in the loading of the DCC complex seen in *met-2; xol-1* mutants is due to the MET-2-mediated change in timing of loss of developmental plasticity.

## Materials and methods

### Worm maintenance

All strains were maintained on NG agar plates with *E. coli* (OP50) as a food source, using standard methods [41]. Strains include: N2 Bristol strain (wild type); TY4403 *him-8 (e1489)* IV; TY1807 *xol-1 (y9)* X; WM458 *xol-1 (ne4472)* X; MT13293 *met-2 (n4256)* III; EKM261 *met-2 (n4256)* III; *xol-1 (ne4472)* X; EKM245 *TY1::CeGFP::FLAG::sdc-2*; *xol-1 (ne4472)* X, EKM84 *TY1::CeGFP::FLAG::sdc-2*; EL634 *3xFLAG::met-2* III; EKM250 *3xFLAG::met-2* III; *xol-1 (ne4472)* X. The *TY1::CeGFP::FLAG::sdc-2* strain was generated by microparticle bombardment with a fosmid clone containing the TY1::CeGFP::FLAG tag using a protocol described in Praitis et al., 2001 [74]. The fosmid clone used for the generation of this strain, fosmid 9660655908490777 F08, was obtained from the TransgeneOme project [75]. Mixed hermaphrodite and male embryos were obtained from *him-8 (e1489)* hermaphrodites. Mutations in the *him-8* gene cause X chromosome nondisjunction in meiosis and result in 38% of progeny being XO males [53,54]. Some strains were provided by the CGC, which is funded by NIH Office of Research Infrastructure Programs (P40 OD010440).

### RNA extraction

Synchronized gravid adult worms were bleached to obtain early embryos. To obtain late embryos, bleached early embryos from synchronized worms were incubated in 1x M9 for 3 hours. For L1 larvae, synchronized early embryos were incubated in 1x M9 with shaking for 24h, then the newly hatched L1 larvae were incubated in OP50 for 3 hours. Samples were lysed by repeated freeze-thaw cycles using liquid nitrogen. TRIzol-chloroform (Invitrogen catalog number 15596026, Fisher Scientific catalog number BP1145-1) separation of the samples was followed by total RNA extraction using the QIAGEN RNeasy Mini Kit (Qiagen catalog number 74104) with on-column DNase I digestion using RQ1 RNase-Free DNase (Promega catalog number M6101). Three replicates were used for each condition for all RT-qPCR experiments. Three replicates were used for early embryo RNA-seq, and four replicates were used for L1 larval RNA-seq.

### RT-qPCR

cDNA was generated from extracted RNA using random hexamers with SuperScript III First-Strand Synthesis System (Invitrogen catalog number 18080051). RT-qPCR reaction mix was prepared using Power SYBR Green PCR Master Mix (Applied Biosystems catalog number 43-676-59) with 10 μl SYBR master mix, 0.8 μl of 10 μM primer mix, 2 μl sample cDNA, and 7.2 μl H2O. Samples were run on the Bio-Rad CFX Connect Real-Time System. Log2-fold change was calculated relative to control (*cdc-42*). Control primers were taken from Hoogewijs et. al. (2008) [76]. Three replicates were analyzed for each genotype. Statistical significance was calculated using 2-tailed unpaired Welch's t-test with unequal variance.

Primer sequences: *met-2* (forward) GAAGCACCGAATCCATTGGC, *met-2* (reverse) CATGTGCTTCTTGTGCGCTG, control (forward) CTGCTGGACAGGAAGATTACG, control (reverse) CTCGGACATTCTCGAATGAAG

### mRNA-seq analysis

Poly-A enrichment, library prep and next-generation sequencing was carried out in the Advanced Genomics Core at the University of Michigan. RNA was assessed for quality using the TapeStation or Bioanalyzer (Agilent). Samples with RINs (RNA Integrity Numbers) of 8 or greater were subjected to Poly-A enrichment using the NEBNext Poly(A) mRNA Magnetic

Isolation Module (NEB catalog number E7490). NEBNext Ultra II Directional RNA Library Prep Kit for Illumina (catalog number E7760L), and NEBNext Multiplex Oligos for Illumina Unique dual (catalog number E6448S) were then used for library prep. The mRNA was fragmented and copied into first strand cDNA using reverse transcriptase and random primers. The 3' ends of the cDNA were then adenylated and adapters were ligated. The products were purified and enriched by PCR to create the final cDNA library. Final libraries were checked for quality and quantity by Qubit hsDNA (Thermofisher) and LabChip (Perkin Elmer). The samples were pooled and sequenced on the Illumina NovaSeqX 10B paired-end 150bp, according to manufacturer's recommended protocols. Bcl2fastq2 Conversion Software (Illumina) was used to generate de-multiplexed Fastq files. The reads were trimmed using CutAdapt v2.3 [77]. FastQC v0.11.8 was used to ensure the quality of data [78]. Reads were mapped to the reference genome WBcel235 and read counts were generated using Salmon v1.9.0 [79]. Differential gene expression analysis was performed using DESeq2 v1.42.0 [80]. Downstream analyses were performed using R scripts and packages. Gene set enrichment analysis was performed using GSEA software v4.3.2 [55].

## Viability assays

For embryonic viability, young gravid adult worms were allowed to lay embryos for 6 h. The number of embryos on each plate was counted and after 1 day at 20°C embryonic viability was scored based on the number of embryos remaining on the plate. For larval viability, the same procedure was followed as for embryonic viability and after another 2 days at 20°C, viability was scored based on the number of adult worms on the plate. For calculating embryonic viability, the number of hatched eggs was divided by the total number of eggs laid. For calculating larval viability, the number of surviving adults was divided by the total number of hatched larvae. Statistical significance was evaluated using chi square tests for each comparison of 2 conditions. The null hypothesis was that no significant difference in viability would be found between the conditions compared. The expected "alive" calculation was formulated by taking the summed % viability between the 2 groups being tested and multiplying that proportion by the sample size ($n$) for each corresponding condition. For embryonic viability, between 500 and 1200 embryos were counted for each condition. For larval viability, between 400 and 900 worms were counted for each condition.

## Brood count

L4s were picked onto a new plate and allowed to develop into a young gravid adult at 25°C and lay eggs for 24h. Adult worms were moved every day until they stopped laying eggs. After the worms were removed from a plate, the number of eggs on that plate were counted. For individual worms, the total number of eggs was counted using the sum of eggs from each plate. Chi square tests were used to compared 2 conditions to determine statistical significance. The null hypothesis was no significant difference in brood count between the 2 conditions tested.

## Immunofluorescence staining

Adult worms were bleached to obtain embryos. After washing in 1x M9, embryos were fixed using finney fix solution (2% v/v paraformaldehyde, 18% v/v methanol, 11.2% v/v witch's brew, 1mM EGTA). Pellets were then frozen in -80°C. Samples were thawed and washed 3x for 15mins each in 1x PBST (PBS with 0.1% Triton X). For foci formation assay, the samples were incubated in rabbit anti-DPY-27 antibody (purified antibody) [50] at 1:200 dilution and mouse anti-nuclear pore complex [Mab414] antibody (Abcam ab24609) at 1:1000 dilution.

The following antibodies and dilutions were used to stain for intensity quantification assays: mouse anti-H3K9me2 (Abcam ab1220) at 1:1000 dilution, rabbit anti-H3K9me3 (Active Motif 39161) at 1:2000 dilution, rat anti-HTZ-1 (purified antibody) [81] (1:100 dilution), rabbit anti-H3 (Abcam ab1791) (1:500 dilution), mouse anti-FLAG M2 antibody (Millipore Sigma F1804) (1:1000 dilution). Samples were incubated for 2 hours at RT for foci formation and MET-2 intensity quantification, and overnight at 4°C for histone modification quantification. Samples were washed 3x 15mins each in 1x PBST. Secondary antibodies used were anti-rabbit FITC (Jackson ImmunoResearch Labs 711-095-152) (1:100 dilution), anti-rabbit Cy3 (Jackson ImmunoResearch Labs 711-165-152) (1:100 dilution), anti-mouse FITC (Jackson ImmunoResearch Labs 715-095-150) (1:100 dilution), anti-mouse Cy3 (Jackson ImmunoResearch Labs 715-165-150) (1:100 dilution), anti-rat FITC (Jackson ImmunoResearch Labs 712-095-153) (1:100 dilution). Samples were incubated in secondary antibody for 1hr at RT. Samples were then washed 3x 15mins each in 1x PBST. Slides were then prepared with Vectashield (Vector Labs catalog number H-1000-10).

## Imaging and quantification

Images of staged embryos were taken with a Hamamatsu ORCA-ER digital camera mounted on an Olympus BX61 epi-fluorescence microscope with a motorized Z drive. The 60× APO oil immersion objective was used for all images. Series of Z stack images were collected at 0.2 μm increments for intensity measurements and 1 μm increments for foci formation. All images shown are 2D projection images of these Z stacks. Quantification was conducted in the Slidebook 5 program (Intelligent Imaging Innovations). For intensity measurement, the mask function was used. Segment masks were drawn for the DAPI signal for each nucleus. Within the statistics menu, the mask statistics function was selected with DAPI as the primary mask. The intensity statistics were selected for the analysis. This procedure generated a number for the intensity corresponding to the Cy3, FITC and DAPI channels from which nuclear signal of our protein of interest was calculated using target protein channel intensity/co-stain channel intensity using both mean and sum intensity. The average intensity ratio for all nuclei of a given genotype was calculated and a 2-tailed unpaired Welch's *t*-test was performed to compare the means of each genotype to the appropriate control. 20 nuclei per condition per embryo stage were quantified, with no more than 3 nuclei imaged from any individual embryo. For foci formation, all genotypes were screened in a blinded fashion. Embryos were classified into 4 categories based on the number of nuclei in each embryo that had DPY-27 foci: 0–25% of nuclei have foci formation on the X, 25–50% of nuclei have foci formation, 50–75% of nuclei have foci formation and 75–100% of nuclei have foci formation. For statistical significance, chi square tests were performed for each pair of conditions.

## Embryo time-lapse imaging

Gravid adult worms were dissected in 1x M9 with a 22G x 1-1/2" needles (Fisher Scientific catalog number 14-840-91) to release embryos. 2-cell to 4-cell embryos were transferred to a slide with an agar pad (2.5% agarose) with 5 μl 1x M9 and a coverslip was placed on top. Embryos were imaged using a Leica MZ16F Stereo Dissecting Microscope mounted with a Nikon DS-Fi3 camera. Embryos were imaged for 6–7 hours taking images at intervals of at least 2 mins with an 11.5x objective lens. Nikon NIS-Elements D v5.41.03 software was used to capture automated time-lapse series. 2-tailed unpaired Welch's t-test with unequal variance was used to calculate statistical significance. 5–10 embryos were imaged for each genotype.

## Supporting information

**S1 Fig.** (A) Boxplot depicting the median log2 fold change in *xol-1* vs WT for genes enriched in L1 larvae. P-value determined by wilcoxon rank-sum test. L1 larval dataset obtained from Spencer et. al. (2011) [44]. Asterisks indicate level of statistical significance (* $p<0.05$; ** $p<0.005$; *** $p<0.001$, n.s not significant).
(TIF)

**S2 Fig.** (A-B) Representative images of SDC-2 loading assay quantified (A) Fig 3F, 50–100 cell embryos and (B) Fig 3G, 20–50 cell embryos using *sdc-2::TY1::CeGFP::FLAG* and *sdc-2::TY1::CeGFP::FLAG; xol-1* embryos. Staining against the nuclear pore complex (NPC) was used as an internal control. (C) SDC-2/HTZ-1 intensity quantification of staged N2 and *xol-1* embryos at 50–100 cells. P-value determined by Welch's t-test with two-tailed distribution and unequal variance. $p = 2.5 \times 10^{-10}$. 20 embryos were quantified in total for each genotype. Asterisks indicate level of statistical significance (* $p<0.05$; ** $p<0.005$; *** $p<0.001$, n.s not significant).
(TIF)

**S3 Fig.** (A-B) Boxplot showing median log2 fold change in *xol-1* vs WT for (A) hermaphrodite-biased gene set without correction for accelerated embryonic development, $p < 2.2 \times 10^{-16}$ or (B) male-biased gene set without correction for accelerated embryonic development, $p = 0.0017$. p-values obtained from wilcoxon rank-sum test. (C) Gene set enrichment analysis on *xol-1* vs WT dataset using sex-biased gene sets (hermaphrodite-biased, $p < 0.001$; male-biased, $p = 0.023$). (D) Log2 fold change in expression of *sdc-1*, *sdc-2* and *sdc-3* from *xol-1*/WT RNA-seq dataset. (E) Log2 fold change in expression of *sdc-1*, *sdc-2* and *sdc-3* in *him-8*/WT RNA-seq dataset. (D-E) Error bars indicate lfcSE standard error. Asterisks indicate level of statistical significance (* $p<0.05$, ** $p<0.005$, *** $p<0.001$, n.s not significant).
(TIF)

**S4 Fig.** (A) RT-qPCR for *met-2* transcripts in *xol-1* mutant and WT late embryos. Error bars indicate standard deviation. Asterisks indicate level of statistical significance (* $p<0.05$, ** $p<0.005$, *** $p<0.001$, n.s not significant). (B) Normalized transcript counts (dcpm) of H3K9me regulators and readers during *C. elegans* embryogenesis. Dataset obtained from Boeck et. al. (2016) [42]. (C) Distribution of frequencies of DPY-27 foci formation in N2 and *met-2* embryos at 50–100 cell stages. P-value determined by chi-square test. (D) Representative images from experiment quantified in (C). Embryos scored: N2 = 48, *xol-1* = 52.
(TIF)

## Acknowledgments

We thank the Mello lab for sharing their *xol-1 (ne4472)* strain, as well as Dr. Eleanor Maine and Dr. Bing Yang for sharing their *3xFLAG::met-2* strain with us.

## Author Contributions

**Conceptualization:** Eshna Jash, Györgyi Csankovszki.

**Data curation:** Eshna Jash.

**Formal analysis:** Eshna Jash, Anati Alyaa Azhar, Hector Mendoza, Zoey M. Tan, Halle Nicole Escher, Dalia S. Kaufman.

**Funding acquisition:** Györgyi Csankovszki.

**Investigation:** Eshna Jash, Anati Alyaa Azhar, Hector Mendoza, Zoey M. Tan, Halle Nicole Escher, Dalia S. Kaufman.

**Methodology:** Eshna Jash, Hector Mendoza, Györgyi Csankovszki.

**Project administration:** Györgyi Csankovszki.

**Validation:** Eshna Jash.

**Visualization:** Eshna Jash.

**Writing – original draft:** Eshna Jash, Györgyi Csankovszki.

**Writing – review & editing:** Eshna Jash, Györgyi Csankovszki.

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
