## [Decision Letter · Decision Letter 0]

13 Jun 2024

Dear Dr Csankovszki,

Thank you very much for submitting your Research Article entitled 'XOL-1 regulates developmental timing by modulating the H3K9 landscape in C. elegans early embryos' to PLOS Genetics.

The manuscript was fully evaluated at the editorial level and by independent peer reviewers. The reviewers appreciated the attention to an important topic but identified some concerns that we ask you address in a revised manuscript.

We therefore ask you to modify the manuscript according to the review recommendations. Your revisions should address the specific points made by each reviewer.

Yours sincerely,

Danielle A. Garsin

Academic Editor

PLOS Genetics

Pablo Wappner

Section Editor

PLOS Genetics

Both reviewers are positive about this work, but suggest some minor changes. In particular, I would like attention given to Reviewer #2's comments about the numbers (N) used in the experiments being made clear. If more time is needed the complete certain experiments to have a greater N, please request it of the journal. We look forward to your revision.

Reviewer's Responses to Questions

**Comments to the Authors:**

Reviewer #1: This is a well written paper with extensive introduction that provides background to the system. The main idea is that xol-1 which turns on the male development pathway has a role in delaying hermaphrodite development. This delay is in part due to met2 overexpression, thus may be linked to premature formation of heterochromatin. Although not very mechanistic, this is a nice story and observation that will be of interest to sex determination and embryonic development fields. A few minor comments for revision:

- “known to” repetition in lines 114-116. Could remove one or both.

- in Figure 1C, what is plotted on the y axis? Legend says normalized transcript counts, but it is not clear normalized to what? Thus, one cannot assess what a number on the y axis means.

- grammar in line 377 “the disruption male biased pathways”

- In histone modification change plots of Figure 6 and elsewhere, the authors should consider using violin plots or plots using individual data points to convey the variability of the data better than the bar plots. The bar plots do not provide insight into the spread of the data. In time to develop plots of Figure 7, it will be better to use individual data points, as few embryos are scored.

- the authors suggest that xol-1 is not a transcriptional regulator of SDC proteins but does regulate SDC-2, perhaps at the level of mRNA translation. The authors could analyze SDC-2 foci by immunofluorescence and compare wild type and xol-1 null mutants to test if their null mutant regulate SDC-2 protein level.

Reviewer #2: This paper investigates the role of xol-1 in C. elegans hermaphrodites. The authors show that there is an acceleration of embryogenesis from the 8-cell stage to bean stage by 13 min in xol-1 mutants. This correlates with a large number of genes changed during early embryogenesis but these gene expression changes go away by the L1 stage. XOL-1 represses hermaphrodite sex determination genes, so there should be an up regulation of these genes in mutants. XOL-1 activates male sex determination so there should be an down regulation of these genes in mutants. Both of these are observed. However, this does not lead to any strong phenotypes. For example, there is no difference in viability at either stage. DPY-27 and SDC-2 load onto the X chromosome slightly earlier at the 50-100 cell stage in xol-1 mutants. There is also more nuclear accumulation of MET-2 in xol-1 mutants and this corresponds with a decrease in H3K9me2 and a later accumulation of H3K9me3, consistent with K9me2 being converted to K9me3 at a higher rate in xol-1 mutants. Consistent with this, the DCC loading acceleration in xol-1 mutants seems to be partially dependent upon met-2. Furthermore, in the met-2; xol-1 double the acceleration of embryogenesis is reverted, but this reversion is well past the met-2 single mutant (which is normal) suggesting there is a genetic interaction more than there is simply a dependence.

Main comments

The paper was very clearly written and easy to read. Overall, the results that the authors observe are quite subtle, indicating that xol-1 does not have a very large function in hermaphrodites. But, on the other hand, the results are internally consistent, suggesting they are likely to be real. In the end, I believe the results are interesting and novel enough to warrant publication. However, with the changes that the authors observe being quite subtle, my main concern is whether the numbers that were assayed were large enough.

The acceleration difference across 4.5hrs is only 13 min. I realize that this is statistically significant, but the N is only >6. I think this needs to be done on far greater numbers of embryos to make the result stronger.

The gene expression changes are consistent with xol-1 embryos being developmentally accelerated also, but again the changes are really small. For the RNAseq in figures 1 and 4, there is no N indicated for number of replicates. With the small gene expression changes, the number of replicates should be at least 3 and perhaps 5.

There are no N for the number of embryos that were scored in figure 3.

In figure 5D,E there is no N for number of embryos

In figures 3 and 5, there is no internal staining control for normalization of the signal. This should be added.

In figure 6, again no N for number of embryos

In figure 7A, there is no WT, so it is not possible to know whether met-2 mutants are actually slower to load the DCC complex. This should be added.

In figure 7 there is also no N number of embryos scored

Minor Comments

What stage were the embryos for RNAseq? Perhaps I missed it but I think it just says early embryos.

How was fertilized vs. unfertilized determined?

**Have all data underlying the figures and results presented in the manuscript been provided?**

Reviewer #1: **No: **GSE262626 is private thus not available to reviewer.

Reviewer #2: Yes

PLOS authors have the option to publish the peer review history of their article (what does this mean?). If published, this will include your full peer review and any attached files.

Reviewer #1: No

Reviewer #2: **Yes: **David J. Katz

---

## [Decision Letter · Decision Letter 1]

30 Jul 2024

Dear Dr Csankovszki,

We are pleased to inform you that your manuscript entitled "XOL-1 regulates developmental timing by modulating the H3K9 landscape in C. elegans early embryos" has been editorially accepted for publication in PLOS Genetics. Congratulations!

Yours sincerely,

Danielle A. Garsin

Academic Editor

PLOS Genetics

Pablo Wappner

Section Editor

PLOS Genetics

Comments from the reviewers (if applicable):

Reviewer's Responses to Questions

**Comments to the Authors:**

Reviewer #1: Iitial comments are addressed.

**Have all data underlying the figures and results presented in the manuscript been provided?**

Reviewer #1: None

PLOS authors have the option to publish the peer review history of their article (what does this mean?). If published, this will include your full peer review and any attached files.

Reviewer #1: No

**Data Deposition**

http://datadryad.org/submit?journalID=pgenetics&manu=PGENETICS-D-24-00355R1

**Press Queries**

---

## [Editor Report · Acceptance letter]

10 Aug 2024

PGENETICS-D-24-00355R1 

XOL-1 regulates developmental timing by modulating the H3K9 landscape in C. elegans early embryos 

Dear Dr Csankovszki, 

We are pleased to inform you that your manuscript entitled "XOL-1 regulates developmental timing by modulating the H3K9 landscape in C. elegans early embryos" has been formally accepted for publication in PLOS Genetics! Your manuscript is now with our production department and you will be notified of the publication date in due course.

With kind regards,

Zsofia Freund

PLOS Genetics

On behalf of:
